# Outcome of Robot-Assisted Bilateral Internal Mammary Artery Grafting via Left Pleura in Coronary Bypass Surgery

**DOI:** 10.3390/jcm8040502

**Published:** 2019-04-12

**Authors:** Chieh-Jen Wu, Hsin-Hung Chen, Pei-Wen Cheng, Wen-Hsien Lu, Ching-Jiunn Tseng, Chi-Cheng Lai

**Affiliations:** 1Cardiovascular Center, Kaohsiung Veterans General Hospital, Kaohsiung 813, Taiwan; cjwu@vghks.gov.tw; 2Graduate Institute of Clinical Medicine, College of Medicine, Kaohsiung Medical University, Kaohsiung 807, Taiwan; 3Department of Medical Education and Research, Kaohsiung Veterans General Hospital, Kaohsiung 813, Taiwan; shchen0910@gmail.com (H.-H.C.); peiwen420@gmail.com (P.-W.C.); cjtseng@vghks.gov.tw (C.-J.T.); 4Yuh-Ing Junior College of Health Care & Management, Kaohsiung 821, Taiwan; 5School of Medicine, National Yang-Ming University, Taipei 112, Taiwan; whlu@vghks.gov.tw; 6Department of Pediatrics, Kaohsiung Veterans General Hospital, Kaohsiung 813, Taiwan; 7Department of Cardiology, Kaohsiung Municipal United Hospital, Kaohsiung 804, Taiwan; 8Department of Biological Sciences, National Sun Yat-Sen University, Kaohsiung 804, Taiwan

**Keywords:** bilateral internal mammary artery, coronary artery disease, coronary artery bypass grafting, da Vinci, sternotomy, outcome

## Abstract

Studies are extremely limited for the investigation of the clinical outcome of da Vinci robot-assisted bilateral internal mammary artery (BIMA) grafting in coronary artery bypass grafting (CABG) surgery. This study aimed to explore the short-term outcome of da Vinci robot-assisted BIMA grafting through the left pleural space. Relevant data were collected from patients with multi-vessel coronary artery disease receiving two kinds of CABG: a group of patients receiving da Vinci robot-assisted CABG with BIMA grafting, and another group of patients receiving sternotomy CABG with BIMA grafting. Primary endpoints, which included cardiovascular and renal endpoints, were analyzed between the groups using the chi-square test, analysis of variance test, and Kaplan–Meier analysis. Compared with the conventional group (*n* = 22), the robotic group (*n* = 22) had a significantly longer operation time (12.7 ± 1.7 vs. 8.5 ± 1.5 hours; *p* < 0.01) and a marginally lower mean of serum creatinine at baseline (1.2 ± 0.3 vs. 2.0 ± 1.7 mg/dL; *p* = 0.04). Primary endpoints (5, 22.7% vs. 12, 54.5%; *p* = 0.03) and renal endpoints (1, 4.5% vs. 7, 31.8%; *p* = 0.02) at six months were significantly reduced in the robotic group compared with the conventional group. There were no differences in cardiovascular endpoints at six months between the groups (1, 4.5% vs. 0; *p* = 1.00). The data showed that da Vinci robot-assisted BIMA grafting was safe, with equal cardiovascular events and lowered renal events at six months, as compared to conventional sternotomy BIMA grafting, despite the longer procedure time. The short-term study suggests that da Vinci robot-assisted BIMA grafting may be considered a favorable surgical option for patients with severe coronary artery disease.

## 1. Introduction

International guidelines recommend coronary artery bypass grafting (CABG) surgery as a treatment option in patients with left main disease and/or multi-vessel coronary artery disease (CAD) [1,2,3]. Conventional CABG surgery requires sternotomy, which generates a long mid-sternal wound scar and potentiates sternal wound infection, particularly in patients receiving bilateral internal mammary artery (BIMA) grafting [4,5,6]. On the other hand, numerous studies have documented the clinical benefit of BIMA grafting, including prolonged graft patency [7,8], lowered adverse cardiovascular events [8,9], and improved survival [9,10,11,12,13]. A conflicting result obtained from a registry study showed that BIMA grafting was not associated with better outcomes compared with single internal mammary artery grafting [14]. In order to minimize surgical trauma and avoid sternotomy, a robot-assisted technique using a da Vinci operator system (Intuitive Surgical, Mountain View, CA, USA) was introduced into the field of CABG surgery. A few studies showed that robotic CABG surgeries had favorable cardiovascular outcomes [15,16,17,18]. In addition, a meta-analysis of 16 pooled studies exhibited that there were significantly fewer renal failure events in robot-assisted endoscopic CABG, as compared to conventional CABG [19]. However, no studies have reported on the composite endpoints of cardiovascular and renal events between da Vinci robot-assisted BIMA grafting and conventional BIMA grafting. Therefore, this retrospective study was designed to investigate the primary endpoints of cardiovascular and renal events between the two patient groups with severe CAD receiving robot-assisted BIMA graft via the left pleura or conventional sternotomy BIMA grafting. The present results were expected to elucidate the short-term primary outcome and help guide surgical decision in patients with severe CAD.

## 2. Materials and Methods

### 2.1. Study Design and Patient Selection

The study was a single-center, retrospective, non-randomized, and non-controlled observational study based upon the analysis of the database of a single medical center, and was designed to detect the differences in adverse events between two patient groups: Group 1 of patients receiving da Vinci robot-assisted CABG with BIMA grafting via left pleura and Group 2 of patients receiving conventional sternotomy CABG with BIMA grafting. Consecutively collected patients were those who had been angiographically diagnosed with left main disease and/or multi-vessel CAD, and thereafter received two kinds of CABG surgeries. They were matched by age. For both groups, femoral artery and vein cannulations were performed for the preparation of cardiopulmonary bypass with systemic heparinization. Beating heart CABG with BIMA skeletonization and mobilization was routinely conducted in all patients, with an on- or off-pump according to the patient’s hemodynamic status and the operator’s discretion. A radial artery (RA) or great saphenous vein (GSV) was harvested as a conduit with the right internal mammary artery (RIMA) for anastomosis of the left circumflex artery (LCX) and of the right coronary artery (RCA). The procedures of the two kinds of CABG surgeries are detailed below. Patients were excluded if they received an urgent CABG, received a hybrid of percutaneous coronary intervention (PCI) and CABG, received left internal mammary artery (LIMA) grafting or RIMA grafting alone, presented with ST-segment elevation myocardial infarction, or had previously been enrolled in another clinical study. All patients received a daily dose of 100 mg aspirin indefinitely, in combination with a daily dose 75 mg clopidogrel as dual antiplatelet therapy with an expected duration of at least one year after the index surgeries. Clinical follow-up was scheduled in hospital, three months, and six months after discharge for data collection. Primary endpoints consisted of a composite of cardiovascular and renal endpoints. Relevant data were recorded for patient characteristics, clinical presentations, and adverse events during the follow-up period. All patients in the two groups were thoroughly informed about the procedure preoperatively and provided written consent. The study protocol had been examined and approved by the committee of the hospital. The present study was performed in accordance with the Declaration of Helsinki and local regulatory guidelines.

### 2.2. Da Vinci Robot-Assisted CABG with BIMA Grafting via Left Pleura

Each beating-heart da Vinci robot-assisted CABG with unilateral BIMA grafting was performed with an on- or off-cardiopulmonary pump according to a patient’s hemodynamic status by an experienced team led by a well-trained CABG surgeon with more than 15 years of CABG surgery experience. After general anesthesia and aseptic procedures, a double-lumen was intubated for single right-lung or bilateral low-volume ventilation in order to mobilize the BIMA grafts. A BIMA graft was mobilized through the left pleura using the da Vinci robot-assisted operation system, with the left chest elevated at approximately 30 degrees and with the patient in supine position. The chest cavity was insufflated with carbon dioxide to expand the surgical space. An RIMA was first mobilized through the left pleural and pre-mediastinal space with two-lung low-volume ventilation. Then, an LIMA was mobilized with single right-lung ventilation. The vessels were anastomosed by direct-vision through a left thoracotomy in the second intercostal space with a surgical wound of about 6–12 cm in length. The mobilized LIMA was anastomosed with the left anterior descending artery (LAD); the RA or GSV served as an additional conduit for connection with the RIMA and for anastomoses of the LCX-obtuse marginal branch (LCX-OM) and/or the RCA-posterior descending artery (RCA-PDA).

### 2.3. Conventional Sternotomy CABG with BIMA Grafting

Patients received a full mid-sternotomy for conventional CABG surgeries with BIMA grafting. The mobilized BIMA and harvested RA or GSV in each patient were anastomosed with native coronary arteries by direct-vision through a mid-sternal exposed surgical wound of about 20–30 cm in length. Similar to the da Vinci robot surgery, the RA or GSV served as an additional conduit for connection with RIMA and for anastomoses with LCX-OM and/or RCA-PDA. All patients in the two groups were treated and monitored postoperatively at the intensive care unit.

### 2.4. Definitions

Adverse events in hospitals were defined as wound infection, severe blood loss requiring blood transfusion, pleural effusion, pulmonary edema, and pneumonia. The definition of CAD was coronary stenosis exceeding 50% in diameter of an adjacently normal segment. Estimated glomerular filtration rate (eGFR) was calculated using the modification of diet in renal disease equation. The definition of CKD stages was according to stratified values of eGFR recommended by the guidelines [20]. Consumed units of packed red blood cells or fresh frozen plasma were defined as the sum of the blood units used in hospital. Each operator determined the use of an intra-aortic balloon pump (IABP) according the hemodynamics in the peri-operative period. Myocardial infarction (MI) was defined according to the clinical symptoms, the level of serum troponin I >5 μg/L, new eletrocardiographic changes, or echocardiographic evidence of new regional wall motion abnormality. Operation room time was defined as the time interval between the patient’s arrival at the operation room and the patient’s departure from the recovery room.

### 2.5. Primary Endpoints at Six Months

Primary endpoints in the study included cardiovascular and renal endpoints. Cardiovascular endpoints including all-cause mortality, non-fatal MI, repeated revascularization, and non-fatal hemorrhagic and ischemic stroke; renal endpoints included a rise of serum creatinine >0.5 mg/dL above the baseline value, creatinine doubling (at least a 100% raise from the basal level of serum creatinine), and occurrence of CKD stage 4 or 5. An endpoint event was confirmed by two independent physicians according to the clinical symptoms and signs, laboratory data, electrocardiographic findings, and/or diagnostic images. The follow-up period was six months.

### 2.6. Statistical Analysis

All variables were statistically analyzed using SPSS software version 22 (SPSS Inc., Chicago, IL, USA). All categorical data and rates are displayed as numbers (percentages), and the continuous data are shown as means ± standard deviation. Baseline and outcome data between the groups were compared using chi-square test (*χ*^2^) or Fisher exact test for categorical variables, and the analysis of variance test for continuous variables. Kaplan–Meier analysis with log-rank test was used to detect differences in cumulative event-free survival at six months between the two groups. A *p* value < 0.05 with two-sided 95% confidence interval was considered statistically significant for all tests. Analysis was conducted as time to the first event involving primary endpoints, without double counting of events.

## 3. Results

### 3.1. Patient Demographic and Characteristic Data

Data for a total of 44 patients were collected from November 2010 to January 2016 (Figure 1). In this cohort, 22 patients with left main disease and/or multi-vessel CAD received da Vinci robot-assisted CABG with BIMA grafting through the left pleura. The 22 age-matched patients received conventional sternotomy CABG with BIMA grafting. Figure 2A,B show endoscopically mobilized BIMA and the minimal surgical scars after robot-assisted surgery. Figure 2C,D show a large sternotomy wound and BIMA anastomosed with coronary arteries, and a larger surgical scar after conventional sternotomy surgery. The operation time was significantly longer in robotic CABG as compared with sternotomy CABG (12.7 ± 1.7 vs. 8.5 ± 1.5 h; *p* < 0.01). The mean serum creatinine at baseline was marginally lower in robotic CABG compared with sternotomy CABG (1.2 ± 0.3 vs. 2.0 ± 1.7 mg/dL; *p* = 0.04), whereas the mean eGFR was identical between groups (57.9 ± 31.5 vs. 65.4 ± 13.6 mL/min/1.73 m^2^; *p* = 0.32). The baseline characteristics of the two groups are shown in Table 1.

### 3.2. In-Hospital Adverse Events

In-hospital adverse events did not differ between the groups (*p* > 0.05), except for a significantly lower incidence of CKD stages 4 and 5 in patients who received da Vinci robot-assisted CABG with BIMA grafting (1, 4.5% vs. 10, 45.4%, respectively; *p* < 0.01). In-hospital cardiovascular events were equal between the two groups, including death, non-fatal MI, repeated revascularization, and non-fatal stroke (*p* > 0.05). The adverse events in hospital are outlined in Table 2.

### 3.3. Primary Endpoints at Six Months

Primary endpoints at six months were significantly reduced in robotic CABG compared with sternotomy CABG (5, 22.7% vs. 12, 54.5%; *p* = 0.03) (Table 3). The finding was reinforced by the Kaplan–Meier analysis (*p* = 0.03 by log-rank test) (Figure 3). The significant reduction was mainly contributed by the fewer renal events in robotic surgery (1 vs. 7; *p* = 0.02). The cumulative rates of cardiovascular endpoints at six months were identical, including all-cause mortality (0 vs. 0; *p* = 1.0), non-fatal MI (0 vs. 1; *p* = 1.0), and non-fatal stroke (0 vs. 0; *p* = 1.0).

## 4. Discussion

The study generated three major findings: (1) primary endpoints were significantly reduced in patients receiving da Vinci robot-assisted BIMA grafting via the left pleura at six months, as compared with patients receiving sternotomy CABG with BIMA grafting; (2) the significant reduction in primary endpoints was contributed by the fewer renal events in robotic CABG. Cardiovascular endpoints were equal between the groups; (3) robotic CABG compared with conventional CABG had smaller surgical wounds but longer operation times, and identical hospital and ICU stay. The da Vinci robot-assisted CABG with BIMA grafting via the left pleura is considered a new surgical treatment option for patients with severe CAD.

Robotic CABG had a significant reduction in the rate of primary cardiovascular and renal endpoints as compared to sternotomy CABG. The relatively better renal function at baseline in da Vinci robot-assisted CABG with BIMA grafting may partially account for the significant reduction in adverse renal events in hospital and at six months. Renal dysfunction was reported to independently predict post-CABG adverse events, such as mortality and/or morbidity [21,22,23], and advanced renal diseases [24,25]. Compared with robotic CABG with BIMA grafting, the poorer renal function in conventional CABG possibly resulted in renal deterioration after CABG surgeries. In addition, the finding was consistent with the result obtained from a meta-analysis of 16 pooled studies which showed that robot-assisted CABG versus conventional CABG was associated with fewer renal failure events [19]. Furthermore, renal function may be preserved in off-pump CABG surgeries [26], despite the equality of cardiopulmonary pump use observed between the groups. A shorter elapsed time from coronary angiography until off-pump CABG was also reported to deteriorate renal function [27]. Nevertheless, the reason why there were fewer renal events in robotic CABG versus conventional CABG remains unclear. The finding needs to be further confirmed.

On the other hand, the two groups had identical rates of adverse cardiovascular events in hospital and at six months. The finding implicated that the da Vinci robot-assisted unilateral BIMA grafting was not inferior to the conventional BIMA grafting in terms of short-term cardiovascular outcome. Furthermore, the data displayed satisfactory results with freedom from subsequent wound infection and post-operative ischemic stroke. A few trials indicated that BIMA harvesting may lead to a decline in blood supply to the sternum and increase the risk of sternal wound infection [4,5,6]. A BIMA graft-sparing aortic clamp and aortic anastomosis may reduce the risk of peri-operative ischemic stroke [8,17].

The present results disclosed that robotic CABG compared with conventional CABG had longer operation times but without longer ICU and hospital stays, and ventilation times. Similar to conventional BIMA mobilization, robot-assisted BIMA mobilization requires a longer procedure time [28,29], especially in RIMA mobilization through the left pleural and pre-mediastinal space. Unilateral BIMA grafting minimizes the surgical wounds but prolongs the operation times. The robot-assisted operation time may be shortened through collaborative team-work in a learning-curve manner [28]. This will increase operator comfort, reduce fatigue, and allow patients receiving da Vinci robot-assisted CABG with BIMA grafting to have safer interventional procedures, therefore greatly improving patient outcomes [30].

Several limitations have to be emphasized: (1) as the study had few case numbers, low event rates, short follow-up, and primary selection bias, its results should be interpreted with caution; (2) the heterogeneity at baseline and unmeasured confounders between the two different surgeries may have affected the outcomes; (3) other factors, such as hemodynamics, angiographically-proven coronary lesions, and systolic function at baseline, that differed between the two groups could have potentially influenced the outcomes, as these were not investigated in the study; (4) in the study, it was not mandatory to routinely check coronary angiograms to survey the graft patency.

In conclusion, the data showed that da Vinci robot-assisted CABG with left-pleural BIMA grafting compared with conventional sternotomy CABG with BIMA grafting did not increase in-hospital and short-term cardiovascular events but possibly lowered adverse renal events. The study suggests that the robot-assisted CABG with left-pleural BIMA grafting can be considered a new surgical option for patients with severe CAD. Further larger studies are needed to investigate long-term outcomes.

## Figures and Tables

**Figure 1 jcm-08-00502-f001:**
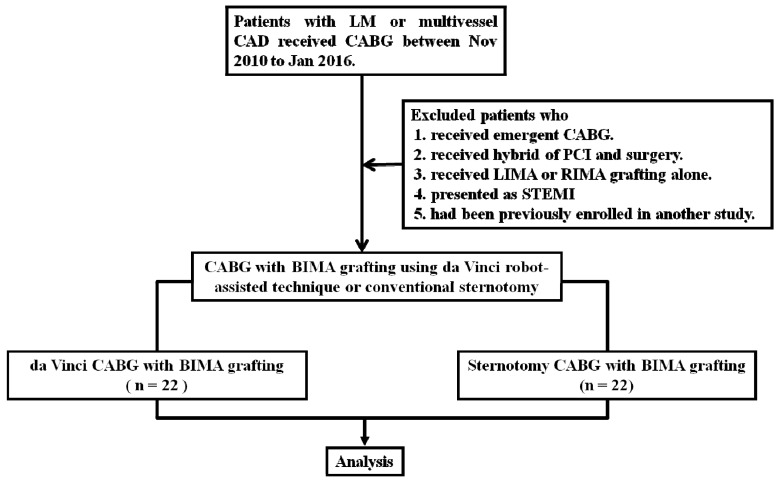
The patient flow chart. LM = left main; CABG = coronary artery bypass grafting; CAD = coronary artery disease; PCI = percutaneous coronary intervention; LIMA = left internal mammary artery; RIMA = right internal mammary artery; STEMI = ST-segment elevation myocardial infarction; BIMA = bilateral internal mammary artery.

**Figure 2 jcm-08-00502-f002:**
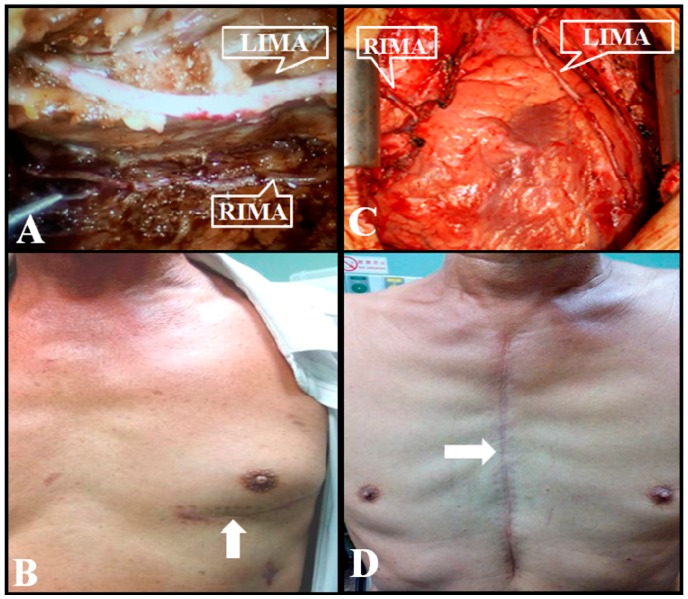
Wound healing in robotic and conventional sternotomy-assisted bilateral internal mammary artery grafting via the left pleura in coronary artery bypass grafting surgery. (**A**) Endoscopy shows that the left internal mammary artery (LIMA) (upper) and the right internal mammary artery (RIMA) (lower) were mobilized using the da Vinci operator system. (**B**) Surgical wounds (white arrows) of da Vinci robot-assisted CABG are small. (**C**) Finished anastomoses of LIMA (right) and RIMA (left) with coronary arteries are displayed in an explored sternal area of a sternotomy CABG. (**D**) A long mid-sternal wound scar (white arrow) is shown in a patient who had received a sternotomy CABG.

**Figure 3 jcm-08-00502-f003:**
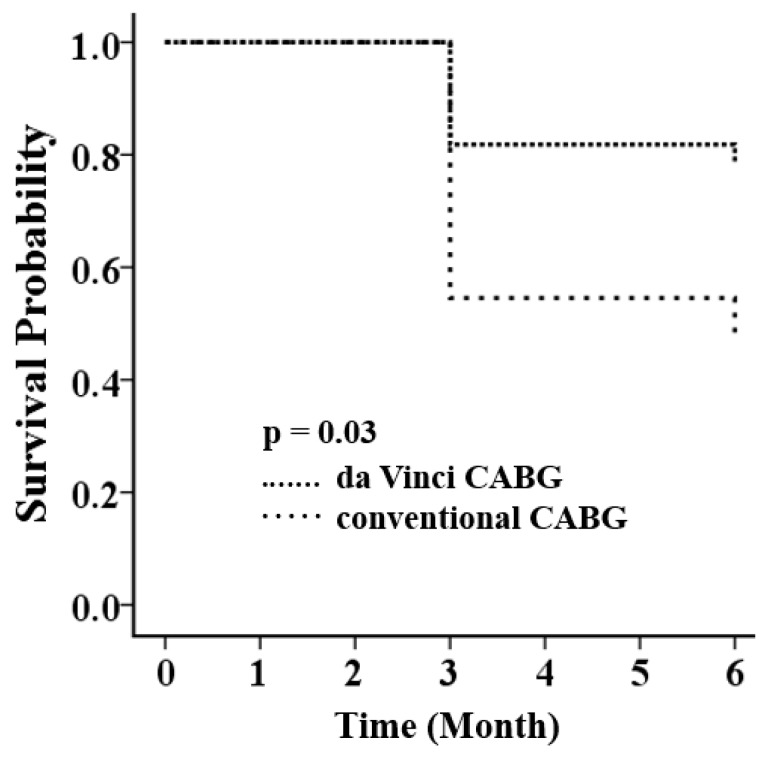
Kaplan–Meier survival analysis shows a significant reduction in the cumulative incidence of the composite of cardiovascular and renal endpoints at six months (*p* = 0.03 by log-rank test), including all-cause mortality, non-fatal myocardial infarction, repeated revascularization, non-fatal stroke, the presence of chronic kidney disease stage 4 or 5, creatinine doubling, and a raise of the baseline creatinine >0.5 mg/dL in patients receiving da Vinci CABG as compared with those receiving conventional CABG. CABG = coronary artery bypass grafting.

**Table 1 jcm-08-00502-t001:** Baseline characteristics between two surgical modalities.

	Da Vinci Robotic Surgery (*n* = 22)	Sternotomy Surgery (*n* = 22)	*p* Value
Male	20 (90.9)	21 (95.5)	1.00
Age (years)	61.2 ± 12.0	62.9 ± 10.5	0.63
Body mass index (kg/m^2^)	26.9 ± 4.0	27.3 ± 3.2	0.70
Risk factors for CAD			
Diabetes mellitus	11 (50.0)	10 (45.5)	1.00
Hypertension	17 (77.3)	18 (81.8)	1.00
Dyslipidemia	13 (59.1)	12 (54.5)	1.00
Hyperuricemia	1 (4.5)	0 (0)	1.00
Drug allergy	5 (22.7)	1 (4.5)	0.19
Alcohol drinking	6 (23.7)	1 (4.5)	0.95
Cigarette smoking	10 (45.5)	11 (50.0)	1.00
Family history of CVD	8 (36.4)	2 (9.1)	0.03
CAD condition			
LM	10 (45.5)	13 (59.1)	0.55
TVD	17 (77.3)	20 (90.9)	0.41
DVD	5 (22.7)	2 (9.1)	0.41
Grafting conduit			
BIMA	22 (100.0)	22 (100.0)	1.00
Radial artery	19 (86.4)	19 (86.4)	1.00
GSV	3 (13.6)	3 (13.6)	1.00
Anastomosis number	3.0 ± 0.6	3.4 ± 0.7	0.34
On pump	10 (45.5)	16 (72.7)	0.12
On pump time (min)	123.9 ± 96.2	131.4 ± 59.8	0.81
Operation room time (h)	12.7 ± 1.7	8.5 ± 1.5	<0.01
Biomarkers at admission			
Hemoglobin (g/dL)	13.7 ± 1.6	12.7 ± 2.4	0.11
Creatinine (mg/dL)	1.2 ± 0.3	2.0 ± 1.7	0.04
eGFR (mL/min/1.73 m^2^)	65.4 ± 13.6	57.9 ± 31.5	0.32
CKD stage			0.11
I/II	15 (68.2)	10 (45.5)	
III	7 (31.8)	7 (31.8)	
IV	0 (0)	3 (13.6)	
V	0 (0)	2 (9.1)	

Continuous data are presented as mean ± standard deviation; category data are presented as number (percentage); CABG = coronary artery bypass grafting; BMI = body mass index; CAD = coronary artery disease; CVD = cardiovascular disease; LM = left main; TVD = triple vessel disease; DVD = double vessel disease; LIMA = left internal mammary artery; RIMA = right internal mammary artery; eGFR = estimated glomerular filtration rate; CKD = chronic kidney disease.

**Table 2 jcm-08-00502-t002:** Adverse events in hospital between two surgical modalities.

	Da Vinci Robotic Surgery (*n* = 22)	Sternotomy Surgery (*n* = 22)	*p* Value
Hospital stay (days)	21.0 ± 8.8	24.4 ± 14.0	0.34
ICU stay (days)	4.8 ± 3.5	5.0 ± 3.3	0.90
Ventilator weaning (days)	2.2 ± 1.8	2.3 ± 2.3	0.94
Renal events			
CKD stages IV/V	1 (4.5)	10 (22.7)	<0.01
Creatinine change >0.5 mg/dL	5 (22.7)	7 (31.8)	0.50
Doubling creatinine (mg/dL)	1 (4.8)	3 (13.6)	0.61
eGFR (mL/min/1.73 m^2^) *	58.5 ± 22.1	45.5 ± 31.6	0.12
Hemodialysis	0 (0)	1 (4.5)	1.00
The lowest hemoglobin (g/dL)	11.9 ± 1.5	12.1 ± 1.5	0.57
Blood transfusion (U)			
FFP	5.3 ± 3.8	3.8 ± 4.3	0.28
PRBC	2.1 ± 2.0	1.4 ± 2.2	0.30
Adverse events	4 (18.2)	5 (27.8)	0.70
Death	0 (0)	0 (0)	1.00
Myocardial infarction	0 (0)	0 (0)	1.00
Stroke	0 (0)	0 (0)	1.00
Wound infection	1 (4.5)	1 (5.6)	1.00
Pneumonia	1 (4.5)	0 (0)	1.00
Pleural effusion	1 (4.5)	2 (11.1)	1.00
Post-operation IABP	2 (9.1)	3 (13.6)	1.00

Continuous data are presented as mean ± standard deviation; category data are presented as number (percentage); CABG = coronary artery bypass grafting; CKD = chronic kidney disease; eGFR = estimated glomerular filtration rate; FFP = fresh frozen plasma; PRBC = packed red blood cells; ICU = intensive care unit; eGFR * = indicate the lowest value of eGFR in hospital; IABP = intra-aortic balloon pump.

**Table 3 jcm-08-00502-t003:** Clinical outcomes at six months between two surgical modalities.

	Da Vinci Robotic Surgery (*n* = 22)	Sternotomy Surgery (*n* = 22)	*p* Value
Finished six months follow-up	19 (86.5)	21 (95.5)	
Primary endpoints	5 (22.7)	12 (54.5)	0.03
Cardiovascular events	1 (4.5)	0 (0)	1.00
Mortality	0 (0)	0 (0)	1.00
Myocardial infarction	1 (4.5)	0 (0)	1.00
Stoke	0 (0)	0 (0)	1.00
Renal events	1 (4.5)	7 (31.8)	0.02
CKD stages IV/V	0 (0)	7 (31.8)	0.02
Creatinine doubling	0 (0)	1 (4.5)	1.00
Creatinine change >0.5 mg/dL	1 (4.5)	5 (22.7)	0.18
Hemodialysis	0 (0)	1 (4.5)	1.00
eGFR (mL/min/1.73 m^2^)	72.1 ± 19.0	56.9 ± 34.0	0.08
Re-hospitalization *	4 (18.2)	6 (27.3)	0.47

Continuous data are presented as mean ± standard deviation; category data are presented as number (percentage); CABG = coronary artery bypass grafting; * = re-hospitalizations due to any cause; primary endpoint = all-cause mortality, myocardial infarction, stock, CKD stages IV/V, and creatinine change >0.5 mg/dL.

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
