# Peer review of "Outcome of Robot-Assisted Bilateral Internal Mammary Artery Grafting via Left Pleura in Coronary Bypass Surgery"

_jcm, 2019, doi:10.3390/jcm8040502_

Round 1
Reviewer 1 Report
It is indeed interesting to read the results of robotic assisted CABG. However, the statistically significant results were found regarding the hours you need to complete such an operation and this could be potentially concerning if the patient is on by pass.
I read thoroughly your article and I have made some comments or need further clarifications as below:
As per line 80 did you do femoral cannulation even in patients who had full srernotomy? Is this a standard practise at your hospital?
Line 83-85: Radial and SVG how were they used?
Line 86 emergent? You mean Urgent?
Line 91: do you regularly follow up your patients post CABG in 3 and 6 months? Is this retrospective study?
110-111 and 116-117: Radial and SVG anastomosed end to end with RIMA? To provide longer conduit? And were they anastomosed to OM and PDA then?
126: Intra Aortic Ballon Pump for IABP
136: How could you confirm from 2 physicians your endpoints in a retrospective study?
Video showing how you mobilise the heart through small incision would be very interestingAny particular reason that you have mostly men in your study? Just demographics?
Noted that with sternotomy you performed more grafts as per your tables.
Could you please define what do you include in operation room time as it seems too long even for conventional surgery?
183: please calculate having in mind that you have enrolled 5 patients with CKD IV and V in the conventional surgery group
As per table 2 could you please explain why the patients had such long mean stays at hospital>21 days?
13.5% of your patients who had Robot assisted BIMAs did not finish follow up
Another paremeter that could affect renal function is whether you have done the operation on or off pump. Do you have the data showing the differnce?
I realise that you are trying to make use of DaVinci more appealing which could be as the scars are much less and you avoid sternotomy. However, the time needed to complete the operation is significantly higher and particularly if the patient is on bypass could lead to unfavourable results including CKD. It may be attributed to the learning curve - which is fare enough. It may be worth to organise a prospective study in the near future to further examine the results. In this way you will probably lose less patients in follow up and you can standardise your patients.
Author Response
Dear Editors and Reviewers:
On behalf of all authors participating in this study, I have to greatly appreciate your professional review work of our manuscript entitled “Outcome of robot assisted bilateral internal mammary artery grafting via left pleura in coronary bypass surgery” (manuscript number: JCM-16-461935). Your constructive comments help us a great deal in refining the manuscript and your kind decision encourages us to go forward in the future. I here reply to all comments you professionally made in a point-to-point manner as follows:
Reviewer Comments:
It is indeed interesting to read the results of robotic assisted CABG. However, the statistically significant results were found regarding the hours you need to complete such an operation and this could be potentially concerning if the patient is on bypass. I read thoroughly your article and I have made some comments or need further clarifications as below:
Question 1:
As per line 80 did you do femoral cannulation even in patients who had full sternotomy? Is this a standard practice at your hospital?
Reply to Q1:
Thanks a lot for your critical questions. The operator selected the patients who had unstable vital signs, markedly reduced ejection fractions and high risk in both groups to do femoral cannulation for preparation of heart-lung support. So, this was not a routine for all patients receiving full sternotomy but a standard practice for some selected patients who had a high risk of hemodynamic instability during the operation in a need of heart-lung support.
Question 2:
Line 83-85: Radial and SVG how were they used.
Reply to Q2:
We appreciate you for reminding us the key point. We detailed how RA and SVG were used in line 110-111 “;the RA or GSV was severed as a conduit for connection with RIMA and for anastomoses of LCX and RCA.” And in line 117-118 “The same as da Vinci robot surgery, the RA or GSV was served as a conduit for connection with RIMA and for anastomoses with LCX and RCA.” In addition, we need to revise the part of “LCX and RCA” in the two sentences to “LCX and/or RCA.”
Question 3:
Line 86 emergent? You mean Urgent?
Reply to Q3:
Thanks a lot for your kind suggestion. We revised the word of “emergent” to “urgent” in line 86.
Question 4:
Line 110-111 and 116-117: Radial and SVG anastomosed end to end with RIMA? To provide longer conduit? And were they anastomosed to OM and PDA then?
Reply to Q4:
We heartily appreciate your critical questions. The answers are all “Yes” for the three questions. First, RA and SVG as additional longer conduits were anastomosed end-to-end with RIMA. Then, the other ends of RA and SVG were anastomosed with LCX-OM and/or RCA-PDA. Thus, we revised the part in line 110-111 as “;the RA or GSV was severed as an additional conduit for connection with RIMA and for anastomoses of LCX-obtuse marginal branch (LCX-OM) and/or RCA-posterior descending artery (RCA-PDA).”; and in line 117-118 “The same as da Vinci robot surgery, the RA or GSV was served as an additional conduit for connection with RIMA and for anastomoses with LCX-OM and RCA-PDA.”
Question 5:
Line 126: Intra Aortic Ballon Pump for IABP
Reply to Q5:
We apologize for this mistake. Yes, IABP means intra-aortic balloon pump. We revised this as “intra-aortic balloon pump (IABP)” in line 126.
Question 6:
Line 136: How could you confirm from 2 physicians your endpoints in a retrospective study?
Reply to Q6:
Thank you for the critical question. We detail the procedure of confirmation of study endpoints in the study here. First, a research assistant disconnected the row data with the patients’ identifications. He provided the clinical data regarding study endpoints for confirmation of endpoints to two independent physicians. The physicians did not know which surgery group presented with the relevant data and were requested to confirm an endpoint event according to the clinical symptoms and signs, laboratory data, electrocardiographic findings, and/or diagnostic images. The third physicians had to be required for the final result of endpoint confirmation in case of inconsistent decisions between two physicians. Finally, the assistant gathered the results and reconnected the data with patients’ identifications. The endpoints were then belonged to each of the two surgery groups. The prepared data were then statistically analyzed. I hope the detail is helpful to clarify your question.
Question 7:
Video showing how you mobilise the heart through small incision would be very interesting. Any particular reason that you have mostly men in your study? Just demographics?
Reply to Q7:
We are encouraged you are very interested in the mobilization of BIMA through small incision. This is an essential part to minimize the surgical wounds with the robot-assisted technique compared with the conventional sternotomy. In addition, I totally agree with your point and I also think it is just demographic that collected men were predominant.
Question 8:
Noted that with sternotomy you performed more grafts as per your tables.
Reply to Q8:
More grafts with sternotomy were considered by chance. Although their difference did not reach a statistical significance (p = 0.34 > 0.05), the graft number was insignificantly increased in the sternotomy group compared with robot-assisted group.
Question 9:
Could you please define what do you include in operation room time as it seems too long even for conventional surgery?
Reply to Q9:
Thanks a lot for your constructive comment. We add the definition of operation room time in the end of “2.4. Definitions” paragraph as “Operation room time was defined as the time duration between patient’s arrival at operation room and patient’s leave from a recovery room.” in line 129. The definition of operation room time might partially result in the longer operation time in both groups.
Question 10:
183: please calculate having in mind that you have enrolled 5 patients with CKD IV and V in the conventional surgery group.
Reply to Q10:
The reviewer pointed the vital part about 5 patients with CKD IV (3 patients) and CKD V (2 patients) at baseline in the conventional surgery group. The worse renal condition at baseline might partially contribute to the poorer renal outcome. However, additional two events of CKD IV/V developed in the conventional surgery group. The renal events of creatinine change > 0.5 mg/dL occurred in 5 patients and creatinine doubling in 1 patient within 6 months, even the events of endpoints were repeatedly counted. Additionally, the conventional sternotomy group had a trend of association with lower eGFR. In contrast, only one renal event (creatinine change > 0.5 mg/dL) occurred within 6 months in the robot-assisted group. To sum up, renal outcome in the robot-assisted surgery group seems better than that in the conventional sternotomy group. Because many factors including measured and unmeasured variables potentially affected renal outcome, we should avoid over-interpretation of the results. Importantly, one purpose of the study was to investigate safety of the robot-assisted CABG surgery. Our data at least showed the renal outcome of the robot-assisted operation was not inferior to that of the conventional group. The robot assisted technique with BIMA grafting may be considered an option for CABG surgery.
Question 11:
As per table 2 could you please explain why the patients had such long mean stays at hospital>21 days?
Reply to Q11:
We explain the reason why the patients stayed at hospital > 21 mean days as follows. First, our healthcare system (National Health Insurance) support the individual protocol of CABG surgery, including the relatively longer hospital stay for preparation and post-operation care and rehabilitation as did at our medical center. The system will reimburse all fees regarding a CABG surgery during hospitalization even relatively longer hospital stay. It was not a routine to transfer patients to other hospitals for post-CABG care in the system. Second, because some patients who lived far away our hospital were referred for CABG surgery, a series of pre-operation examinations, and explanation of operation risk and complications to patients and their family might spend a few days before surgery. It seemed convenient for such patients to be followed at the other hospitals near they lived after completely recovery.
Question 12:
13.5% of your patients who had Robot assisted BIMAs did not finish follow up
Reply to Q12:
As mentioned above, some referred patients lived far away from the medical center. A few patients went back to home at the other cities and were followed at the other hospitals. Thus, parts of the clinical data were unavailable and uncompleted even we contacted with them by phone with an intention to understand their cardiovascular and renal conditions. Unfortunately, some data such as lab biomarkers were not available during the study period in three cases with robot assisted BIMA operation. So, they did not finish the study follow-up.
Question 13:
Another paremeter that could affect renal function is whether you have done the operation on or off pump. Do you have the data showing the differnce?
Reply to Q13:
Thanks a lot for the critical question. We know several factors, such as the factor of on pump or off pump, have the potential to affect renal function in patients receiving CABG surgery. First, the study did not focus on the impact of on and off pump on renal outcome. Second, several variables and confounders potentially affecting renal outcome should be additionally taken into account for investigation. Third, the statistical power might be declined in this small study population. Therefore, we did not have convincing data showing the difference between groups. That is, it should be underpowered to present the difference in renal outcome in this small study population size with multiple confounders, although the numbers of on pump surgeries were identical between two groups.
Question 14:
I realise that you are trying to make use of DaVinci more appealing which could be as the scars are much less and you avoid sternotomy. However, the time needed to complete the operation is significantly higher and particularly if the patient is on bypass could lead to unfavourable results including CKD. It may be attributed to the learning curve - which is fare enough. It may be worth to organise a prospective study in the near future to further examine the results. In this way you will probably lose less patients in follow up and you can standardise your patients.
Reply to Q14:
We appreciate your understanding about the differences between the two surgeries. The da Vinci robot-assisted surgery provided small surgical scars without sternotomy. The longer time with robot-assisted surgery might be due to the complicated procedure and learning curve. I totally agree at your suggestion that a prospective study may be required to improve study quality and strengthen the present results.
Finally, on behalf of all co-authors, I have to cordially appreciate the Editor and Reviewers again for your professional review work and kind decision to consider the manuscript for publication.
Reviewer 2 Report
Chieh-Jen Wu and coworkers explored short-term outcome of da Vinci robot-assisted BIMA grafting through left pleural space. The research work is well done and present encouraging data in the group of patients receiving da Vinci robot-assisted CABG with BIMA grafting. There is however lack of evidence on the quality of results in the long-term follow-up in terms of by pass re-stenosis. The authors need to document the patency of by-pass and stability of the patients by presenting stress-test data. Furthermore, is there any intraoperative documentation of patency of by-pass in the group of patients receiving da Vinci system?
Author Response
Dear Editors and Reviewers:
On behalf of all authors participating in this study, I have to greatly appreciate your professional review work of our manuscript entitled “Outcome of robot assisted bilateral internal mammary artery grafting via left pleura in coronary bypass surgery” (manuscript number: JCM-16-461935). Your constructive comments help us a great deal in refining the manuscript and your kind decision encourages us to go forward in the future. I here reply to all comments you professionally made in a point-to-point manner as follows:
Reviewer Comments:
Question 1:
Chieh-Jen Wu and coworkers explored short-term outcome of da Vinci robot-assisted BIMA grafting through left pleural space. The research work is well done and present encouraging data in the group of patients receiving da Vinci robot-assisted CABG with BIMA grafting. There is however lack of evidence on the quality of results in the long-term follow-up in terms of bypass re-stenosis. The authors need to document the patency of by-pass and stability of the patients by presenting stress-test data. Furthermore, is there any intraoperative documentation of patency of by-pass in the group of patients receiving da Vinci system?
Reply to Q1:
We heartily appreciate you for the kind comments. Stress tests were not warranted within 6-months follow-up in the study. In general, stress tests might be considered in case of progression of typical chest discomfort within 6 months (symptoms-driven) or after 12-month follow-up (time-driven). So, a stress test was not routinely done for a patient in the study with a short-term follow-up. On the other hand, vascular flows were routinely assessed by Doppler’s flow examinations. An optimal vascular Doppler’s flow was obtained to make sure the intraoperative documentation of patency of bypass in both groups, regardless of robot-assisted group and conventional sternotomy group.
Finally, on behalf of all co-authors, I have to cordially appreciate the Editor and Reviewers again for your professional review work and kind decision to consider the manuscript for publication.
Round 2
Reviewer 1 Report
Thank you for replying satisfactory to my questions. I understand that every hospital can face different difficulties to recruit for a study. It's a good start for your Da Vinci programme. Keep up the good job.
Please make sure you include in your manuscript the explanation regarding hospital stay and theatre time.